# Inter-Rater Agreement in Assessing Risk of Bias in Melanoma Prediction Studies Using the Prediction Model Risk of Bias Assessment Tool (PROBAST): Results from a Controlled Experiment on the Effect of Specific Rater Training

**DOI:** 10.3390/jcm12051976

**Published:** 2023-03-02

**Authors:** Isabelle Kaiser, Annette B. Pfahlberg, Sonja Mathes, Wolfgang Uter, Katharina Diehl, Theresa Steeb, Markus V. Heppt, Olaf Gefeller

**Affiliations:** 1Department of Medical Informatics, Biometry and Epidemiology, Friedrich Alexander University of Erlangen-Nuremberg, 91054 Erlangen, Germany; 2Department of Dermatology and Allergy Biederstein, Faculty of Medicine, Technical University of Munich, 80802 Munich, Germany; 3Department of Dermatology, University Hospital Erlangen, 91054 Erlangen, Germany; 4Comprehensive Cancer Center Erlangen-European Metropolitan Area of Nuremberg (CCC ER-EMN), 91054 Erlangen, Germany

**Keywords:** inter-rater agreement, inter-rater reliability, melanoma, risk of bias, prediction, PROBAST

## Abstract

Assessing the risk of bias (ROB) of studies is an important part of the conduct of systematic reviews and meta-analyses in clinical medicine. Among the many existing ROB tools, the Prediction Model Risk of Bias Assessment Tool (PROBAST) is a rather new instrument specifically designed to assess the ROB of prediction studies. In our study we analyzed the inter-rater reliability (IRR) of PROBAST and the effect of specialized training on the IRR. Six raters independently assessed the risk of bias (ROB) of all melanoma risk prediction studies published until 2021 (n = 42) using the PROBAST instrument. The raters evaluated the ROB of the first 20 studies without any guidance other than the published PROBAST literature. The remaining 22 studies were assessed after receiving customized training and guidance. Gwet’s AC_1_ was used as the primary measure to quantify the pairwise and multi-rater IRR. Depending on the PROBAST domain, results before training showed a slight to moderate IRR (multi-rater AC_1_ ranging from 0.071 to 0.535). After training, the multi-rater AC_1_ ranged from 0.294 to 0.780 with a significant improvement for the overall ROB rating and two of the four domains. The largest net gain was achieved in the overall ROB rating (difference in multi-rater AC_1_: 0.405, 95%-CI 0.149–0.630). In conclusion, without targeted guidance, the IRR of PROBAST is low, questioning its use as an appropriate ROB instrument for prediction studies. Intensive training and guidance manuals with context-specific decision rules are needed to correctly apply and interpret the PROBAST instrument and to ensure consistency of ROB ratings.

## 1. Introduction

Clinical and epidemiological studies devoted to evaluating prognostic and/or risk factors of a specific disease are prone to many forms of bias [1]. Bias is defined as the presence of systematic error in a study that leads to flawed results and thus impairs the validity of study findings [2,3]. To be able to properly interpret study results and to avoid under- or over-estimation of the parameter of interest, it is essential to assess the risk of bias (ROB) of studies [4]. Especially for the appropriate conduct of systematic reviews and meta-analyses, which have become increasingly important in clinical medicine over the last two decades [5], the assessment of the methodological quality of included studies has become a key element and is part of the PRISMA (Preferred Reporting Items for Systematic Reviews and Meta-Analyses) guideline [6]. The need for methodological quality assessments has contributed to the development of a large number of ROB instruments over the last two decades [7,8]. Most of the instruments were developed for specific study designs, such as the revised Cochrane Risk-of-Bias tool (ROB2) for randomized controlled trials [9] or ROBIS (Risk Of Bias In Systematic review) for systematic reviews [10]. Another tool specifically designed for prediction studies and published in 2019 is PROBAST (Prediction Model Risk of Bias Assessment Tool) [11,12]. The development of PROBAST was based on a consensus process consisting of a Delphi procedure involving a panel of 38 experts and a refinement through piloting. The final instrument has a domain-based structure and provides criteria for the evaluation of the methodological quality of studies developing, validating, or updating prediction models [11]. The authors of PROBAST defined bias in the context of predictive studies as “shortcomings in study design, conduct, or analysis [that] lead to systematically distorted estimates of model predictive performance” [2]. Although the tool was published only a few years ago, it has already been used extensively [13]. This demonstrates that PROBAST fills an important gap in the repertoire of ROB tools for predictive studies.

Assessing ROB improves transparency about the methodological quality of studies. However, this is only possible if the ROB instruments themselves are valid and reliable. While validity addresses the extent to which the observed results represent the truth, reliability relates to the extent to which results can be reproduced. Low validity and poor reliability of ROB assessment tools, by impairing the quality of systematic evidence synthesis, may ultimately have an impact on decision-making and quality of patient care [14]. One element of reliability is the inter-rater agreement, which refers to the reproducibility or consistency of decisions between two or more raters [4]. ROB instruments often depend on the experience and personal judgment of raters, which can lead to different ROB ratings when multiple raters assess the same study. Thus, to assess and improve consistency in the application of ROB assessment tools, it is necessary to explore the inter-rater reliability (IRR) of ROB instruments. Up to now, only a few ROB tools have undergone extensive IRR or validity testing by independent groups [15,16,17,18,19]. Overall, these studies revealed deficits in the reliability of the tools examined [15]. There is, however, some evidence that intensive, standardized training for raters may significantly improve the reliability of ROB assessments [14,20].

To the best of our knowledge, hitherto no studies examining the effect of specialized training on the reliability of the PROBAST instrument exist. Therefore, our objectives were (i) to investigate the IRR of this instrument and (ii) to explore the effect of intensive rater training and targeted outcome-specific guidance manuals on the IRR in a representative manner, using melanoma prediction studies as an example.

## 2. Materials and Methods

### 2.1. Study Selection

We included 42 studies reporting development and validation of models predicting the individual risk of melanoma occurrence. The set of studies to be assessed was based on a recent systematic review of melanoma prediction modeling published in 2020 [21] and a literature update performed in August 2021. The update included the forward snowballing technique, which was applied on [21] and two other previously published systematic reviews on the same topic [22,23], and an electronic literature search in PubMed using the same search string as in [21]. Details on the study selection and eligibility criteria were published previously in a report describing the ROB of melanoma prediction studies based on the consensus ROB rating of the rater team [24].

### 2.2. ROB Assessment Using PROBAST

Six raters (I.K., S.M., K.D., T.S., M.V.H., O.G.) assessed independently the ROB of each study using the PROBAST instrument provided on the website [12]. The rater panel was multidisciplinary and consisted of epidemiologists (I.K., O.G.), clinical dermatologists (S.M., M.V.H.), and public health experts (K.D., T.S.) at different levels of professional experience with systematic reviews and ROB assessments. Two raters had no previous experience in this area. Although some of the raters had already performed ROB assessments, none had used the PROBAST instrument before.

PROBAST is structured into four domains: (1) The domain “participants” covers possible sources of bias related to the data sources and the participant selection; (2) the domain “predictors” contains bias through selection and assessment of predictors; (3) the domain “outcome” focuses on possible bias because of definition or determination of the outcome; and (4) the domain “analysis” covers bias linked to estimated predictive performance induced by inappropriate analysis methods or omission of statistical considerations. Each domain was rated individually as either low, high, or unclear. The raters were assisted in judging the ROB for each domain by a total of twenty signaling questions that were answered as yes, probably yes, no, probably no, or no information. Based on the ratings in the four domains, an overall ROB was assigned to each study. According to [11], the overall ROB was obtained by taking the lowest rating of any domain-specific ROB. Thus, a study only received a low overall ROB if all four domains were judged as low.

### 2.3. Rating Process and Training

A timeline of the study is given in Figure 1. Prior to the rating process, an initial meeting was held to discuss the objective and implementation of the ROB assessment. During this meeting, the published PROBAST literature, namely the original PROBAST publication [11] and the explanation and elaboration document [2], was provided to the raters. Thereafter, a random selection of 20 studies was assessed by the raters without any further guidance. After the completion of this part of the rating, two moderated training sessions followed where each PROBAST item was reviewed and its meaning discussed in the group to ensure that all raters interpreted the items in the same way. In addition, disagreements between the raters regarding the ROB ratings of the first twenty studies were discussed in two meetings lasting four hours each to reach consensus decisions. In three cases of sustained disagreement, two independent referees (A.B.P. and W.U.) made the final decisions. A customized guidance manual [24] was developed based on the consensus decisions. It contained decision rules to guide raters in making adjudications for each domain of the PROBAST instrument when specifically applied to melanoma prediction studies, establishing a common standard for the rating process. Afterwards, the ROB of the remaining 22 studies was assessed based on that guidance. Again, six consensus meetings of 1.5 h were held to resolve disagreements regarding the ROB ratings.

### 2.4. Statistical Analysis

We determined the IRR before and after training for the domain-specific and overall ROB ratings. We calculated the pairwise agreement and the agreement at the multi-rater level, respectively. Given that six raters participated in the study, there were fifteen possible pairs of raters. To assess the IRR, we used Gwet’s AC_1_ statistic [25] instead of the better-known kappa statistics. A rationale for this decision detailing the difference between Gwet’s AC_1_ and the kappa statistics can be found in Appendix B. We also reported values of Cohen’s kappa (κ) [26] and Conger’s κ [27] to ensure comparability with other studies. We interpreted an AC_1_ < 0 as poor, 0.0 to 0.20 as slight, 0.21 to 0.40 as fair, 0.41 to 0.60 as moderate, 0.61 to 0.80 as substantial, and 0.81 to 1.00 as almost perfect [28]. Additionally, we calculated pairwise raw agreement proportion before and after training for each PROBAST domain and the overall ROB rating. To quantify the training effect at the multi-rater level, we calculated the difference in agreement between AC_1_ estimates after training and before training (∆AC_1_). We bootstrapped ∆AC_1_ using bias correction and acceleration to obtain 95% confidence intervals (CIs) [29]. Analyses were performed in R version 4.2.1 [30].

## 3. Results

### 3.1. Study Characteristics

The PROBAST assessment of the forty-two studies [31,32,33,34,35,36,37,38,39,40,41,42,43,44,45,46,47,48,49,50,51,52,53,54,55,56,57,58,59,60,61,62,63,64,65,66,67,68,69,70,71,72] resulted in a low overall ROB rating for only one study (2%), while seven studies (17%) received an unclear ROB rating and thirty-four studies (81%) a high ROB rating [24]. The domain “outcome” contributed the highest proportion (n = 37; 88%) of low ROB ratings among all four domains in our investigation. The set of studies before and after training was similar regarding the overall ROB rating. Figure 2 shows the distribution of the overall and domain-specific ROB ratings in the two sets of studies assessed before and after the training. Details of the individual ROB rating results can be found in the Appendix A.

### 3.2. Multi-Rater Agreement

Figure 3 shows the multi-rater agreement before and after training for the four PROBAST domains and the overall ROB rating. Values of AC_1_ before training ranged from 0.071 to 0.535. After training, the agreement ranged from 0.294 to 0.780. The highest agreement was observed in the outcome domain. We observed a significant improvement in the agreement after training compared to before training for the overall ROB rating (∆AC_1_ = 0.405; 95%-CI 0.149–0.630) and the domains “outcome” (∆AC_1_ = 0.245, 95%-CI 0.063–0.595) and “analysis” (∆AC_1_ = 0.194; 95%-CI 0.003–0.365). For the domains “participants” and “predictors”, the improvement in agreement was negligible. The corresponding estimates of Conger’s κ and their difference between before and after training can be found in the Appendix A.

### 3.3. Pairwise Agreement

The distribution of AC_1_ estimates for pairwise agreement before and after training for the domain-specific and overall ROB ratings is shown in the left panel of Figure 4. In addition, the distribution of ∆AC_1_ estimates of pairwise agreement is presented in the right panel of the same figure. The detailed values of all estimates of pairwise agreements can be found in Appendix A. The highest level of agreement, both before and after training, can be found in the domain “outcome”. The median of the differences was greater than 0 for all domains, indicating a positive effect of the training. For the overall ROB rating, the median of ∆AC_1_ was highest (0.427).

The agreement between individual raters and the consensual rating decision before and after training is shown in Table 1. With a few exceptions (n = 5, 17%), agreement with the consensus decision improved across all domains after training for all raters. The amount of improvement varied depending on rater and domain. The highest agreement between raters and consensus decision after training was found in the domain “outcome” (AC_1_ 0.683–0.947).

### 3.4. Comparison of Raw Agreement, Gwet’s AC_1_ and Cohen’s κ for Mean Pairwise Agreement

Table 2 compares the mean pairwise raw agreement, mean AC_1_ and mean Cohen’s κ for all PROBAST domains and the overall ROB rating to ensure the comparability of our results with other studies that did not use the AC_1_ as measure for the IRR. Mean values of the pairwise raw agreement proportion ranged from 0.377 to 0.630 before training and from 0.494 to 0.809 after training with highest values in the domain “outcome”. Due to the adjustment for random agreement between raters, the mean values of the AC_1_ and Cohen’s κ for pairwise agreement were lower than the mean raw agreement, with κ values usually being considerably lower than AC_1_ estimates due to imbalances of marginal distribution of rating results. For the domain “outcome”, where the imbalance of the marginal distribution of rating results was strongest, we observed the highest difference between AC_1_ and Cohen’s κ.

## 4. Discussion

Our results show that without guidance or specific training, the IRR of the PROBAST instrument was low, meaning that the ROB assessment of melanoma prediction studies was not reliable. Training sessions and customized guidance focusing on the implementation of the PROBAST instrument in our particular field of application, namely melanoma prediction studies, significantly improved the agreement for the overall and two domain-specific ROB ratings, which substantiates the need for intensive as well as disease- and study-type-specific training before using the tool.

Slight to moderate agreement was found before training both at two-rater (mean pairwise AC_1_: 0.098–0.534) and at multi-rater level (AC_1_: 0.071–0.535). However, there were substantial differences depending on the domain. In domains requiring high levels of subjective judgment and methodological expertise, such as in the domain “analysis” (mean pairwise AC_1_: 0.142; multi-rater AC_1_: 0.100), agreement was lowest. There was also poor agreement on the overall ROB rating (mean pairwise AC_1_: 0.098, multi-rater AC_1_: 0.071). We observed the highest level of agreement for the domain “outcome” (mean pairwise AC_1_: 0.534, multi-rater AC_1_: 0.535), which is the domain requiring less complex and subjective evaluations than other domains as there is an established definition of the outcome, here cutaneous melanoma, with standard diagnostic procedures that have been used in most studies. Furthermore, our study found that IRR varied widely depending on the pair of raters. The degree of variability was again dependent on the PROBAST domain. Especially for the overall rating, the IRR before training varied strongly across the fifteen different rater pairs (pairwise AC_1_: −0.265–0.873). This clearly demonstrates the subjectivity of the PROBAST instrument and its rater dependency.

To the best of our knowledge, this is the first study assessing the reliability of the PROBAST instrument for prediction studies on a specific outcome. A previous study by Venema et al. [73], which focused on comparing a short form of PROBAST with the full-length PROBAST instrument in their capabilities to identify prediction models for cardiovascular diseases that perform poorly at external validation, also examined the IRR between two reviewers for the ROB assessment on these clinical prediction models. They reported a Cohen’s κ of 0.33, which is in line with our results and allows for the conclusion that the low IRR of the PROBAST tool is not a melanoma-specific problem. Several studies assessed the reliability of other ROB instruments, such as the Cochrane ROB tool and ROBIS. Some of them reported IRRs that were of a similarly low level as in our study [15,17,74,75]. Gates et al. [15] evaluated the IRR of the AMSTAR (A MeaSurement Tool to Assess systematic Reviews), AMSTAR 2, and ROBIS tools. While the IRR for AMSTAR/AMSTAR 2 was in a moderate to good range (AC_1_: 0.5–0.8), the IRR of the ROBIS tool was similar to our results for PROBAST (AC_1_: −0.2–0.6). Könsgen et al. [74] evaluated the IRR of the Cochrane ROB tool using Conger’s κ. Their results for the IRR (0.2–0.5) are slightly higher than our values for PROBAST (Conger’s κ: 0.0262–0.181), but are still in a fair range of agreement. Other studies, including Momen et al. [76] who studied the ROB-SPEO (Studies estimating Prevalence of Exposure to Occupational risk factors) tool, and Hoy et al. [77] who analyzed the IRR of the Hoy tool, report higher IRR estimates (Cohen’s κ: 0.5–0.8 and 0.5–0.9, respectively). However, in both cases the raters were familiar with the use of the tool. They had either been involved in the development of the instrument or received customized guidance before its use, so these IRR values are not comparable to our results before training.

Two possible explanations for the disagreement between raters are conceivable [75]: (i) a relevant piece of information is missed by one or more than one of the raters, (ii) interpretation of the same information is different owing to a subjective component. Training sessions and the development of a targeted and structured guidance manual address the problem of different interpretations of ROB items. Our results after training demonstrated that the IRR of the PROBAST instrument significantly improved in the second part of ROB assessments. At the start of the study, all raters were entirely inexperienced in using PROBAST, so there was a consistent baseline for quantifying the training effect. The largest net gain was achieved in the overall rating (ΔAC_1_: 0.405) and the domain “outcome” (ΔAC_1_: 0.245). When looking at the agreement of rater pairs, it became evident that for the vast majority of the rater pairs, the training improved the IRR. Other researchers have also shown that standardized training leads to a significant improvement in IRR for other ROB instruments [14].

However, high reliability does not imply correctness or validity of the tool. Focusing only on IRR would be insufficient, as high IRR does not necessarily imply that the ratings are correct [78]. Due to the absence of an external gold standard to validate our ROB assessments, we had no choice but to build on our consensus ratings, assuming these to be “correct”. On account of a valid consensus process, where all raters jointly made final decisions, and involvement of two independent referees when no consensus could be reached through discussion, these ratings should be free of individual rater errors and bias. Our results show that, with a few exceptions, training improved agreement with the consensus decision in all domains and for all raters, making us confident that the consensus decisions were correct.

In practical applications comprising ROB assessments, it is not sufficient to simply use the checklist of a published ROB instrument. Specific guidance on how to implement a given instrument to a specific disease condition or study type is essential. Explanations, such as those available for PROBAST [2,13], can help to interpret the items correctly. However, explicit criteria for unclear and high ratings are rarely included, as they depend on the specific application. Therefore, before using the tool, it is important that users conduct training and/or create guidance manuals to address the main methodological problems common in their specific area of research. Valid decision rules for ROB ratings in a given research field require experienced epidemiologists specialized in the area of research that is involved.

Beyond defining decision rules, the rater group will achieve calibration through discussion and develop a common sense of when to apply a low or high ROB rating to a study. Beyond verifiable facts, each rater group develops its own evaluation standard for the ROB classification of studies by means of consensus discussions. Thus, a high IRR is always an indicator of a good calibration within the group. However, a high IRR for a ROB instrument in one rater group does not mean that other rater groups would arrive at the same ROB ratings with the same instrument for the same studies, as it may be that the other raters are “calibrated” differently.

Authors of systematic reviews and meta-analyses are strongly encouraged by guidelines such as PRISMA to incorporate ROB considerations into their process of research synthesis for quality improvement, namely reduction in bias in overall results [6]. However, ROB assessment and interpretation with regard to the strength of evidence assessment will be misleading if based on sub-optimal use of ROB instruments. Our results highlight that raters need to be aware of the limitations of ROB instruments. Detailed guidelines, decision rules, and transparency of the rating process are needed so that readers of systematic reviews can see how the tools were applied and are able to evaluate the results, that is, both the ROB tool and any specific thematic guidance used should ideally be published along with a systematic review.

Due to unbalanced marginal distributions in our ROB ratings, the use of any κ statistic would have potentially underestimated the IRR due to the well-known κ paradox [79,80]. In fact, individual rating categories were often disproportionately represented in some domains of our PROBAST rating. The domain “outcome” was rated as low in 82 out of 120 ratings (68%) before training and in 112 out of 132 ratings (85%) after training. The domains “participants” and “predictors” were rated as low in 63% of the ratings (75/120 and 76/120, respectively) before training. While Gwet’s AC_1_ offered in our case the advantage of addressing the problem of unbalanced marginal rating distributions, it also limits our comparability with other studies as this measure is used less frequently than the more widely used κ statistic. We reduced this limitation by additionally reporting Cohen’s κ for pairwise agreement and Conger’s multi-rater κ for our main results. Additionally, even if Gwet’s AC_1_ is still rather unknown, it has already been used by other researchers for the evaluation of inter-rater agreement [4,15,81].

A limitation to the generalizability of our findings regarding the training effect is that the magnitude of the effect is probably related to how detailed the decision rules were defined. These were developed based on the consensus over the first 20 studies, which evidently could not cover all possible reasons for unclear and high ratings for all domains faced in the remaining 22 studies. This translates to the notion that residual uncertainties will always be an issue arising with future primary literature, necessitating continual update of such customized guidance to original ROB tools. Furthermore, agreement may be higher among raters with a comparably high experience in research methods and epidemiology. The composition of our group was mixed in terms of the field of expertise and experience with systematic reviews and ROB assessments, which may have had some negative impact on IRR results. However, our mixed group of raters likely represents the range of raters that would typically be involved in such activities and thus our results provide a realistic impression of what can be expected from the PROBAST instrument in practice.

## 5. Conclusions

Without targeted guidance, the inter-rater agreement of the PROBAST instrument is low, questioning its use as an appropriate ROB instrument for prediction studies. Therefore, intensive training and guidance manuals with context-specific decision rules for high and unclear ratings are needed to correctly apply and interpret this ROB instrument and to ensure consistency of the ratings.

## Figures and Tables

**Figure 1 jcm-12-01976-f001:**
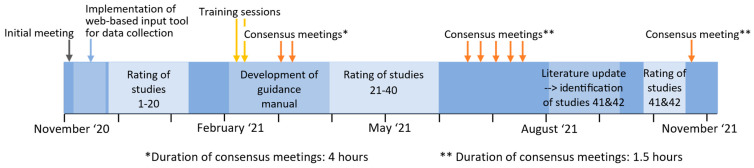
Timeline of the PROBAST assessment study from the initial meeting until the final consensus meeting showing all steps of the study.

**Figure 2 jcm-12-01976-f002:**
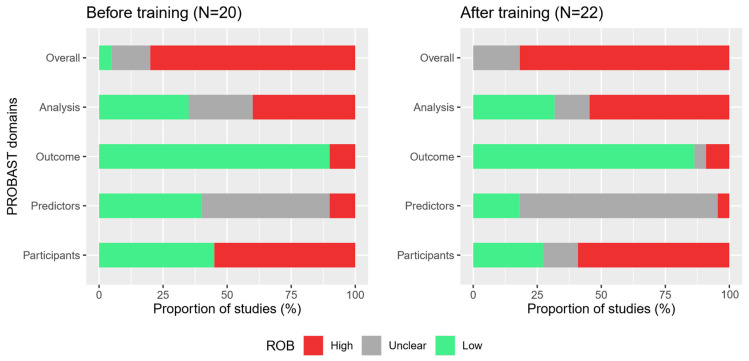
Overall and domain-specific ROB ratings of studies assessed before (n = 20) and after training (n = 22).

**Figure 3 jcm-12-01976-f003:**
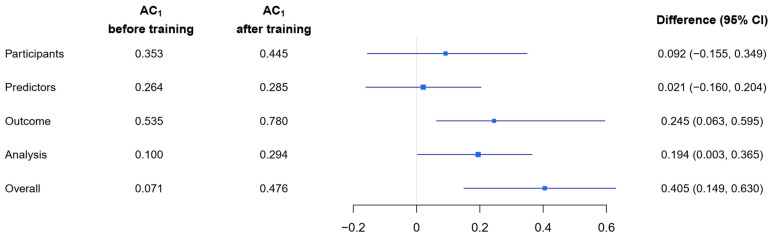
Multi-rater agreement in terms of AC_1_ before and after training for the domain-specific and overall ROB ratings, as well as ∆AC_1_ estimates with bootstrapped 95%-CI.

**Figure 4 jcm-12-01976-f004:**
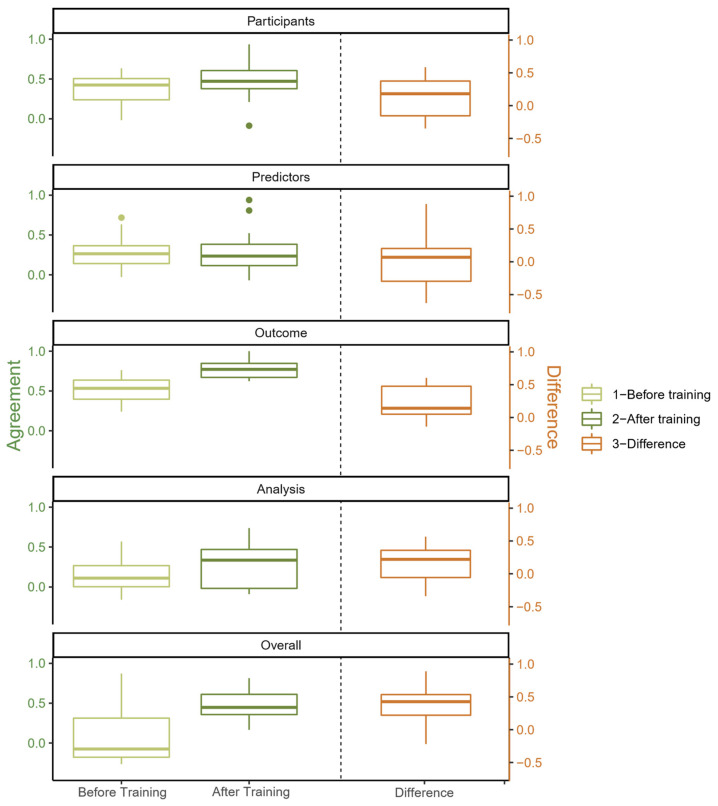
Distribution of pairwise inter-rater agreement in terms of box plots of AC_1_ estimates before and after training for the domain-specific and overall ROB ratings (left part), as well as ∆AC_1_ estimates (right part).

**Table 1 jcm-12-01976-t001:** Agreement in terms of AC_1_ estimates between individual raters and consensus decision before and after training for the domain-specific and overall ROB rating.

	Rater 1	Rater 2	Rater 3	Rater 4	Rater 5	Rater 6
Domain 1: Participants						
Before training	0.730	0.148	0.181	0.173	0.260	0.652
After training	0.675	0.805	0.549	0.546	0.074	0.679
Domain 2: Predictors						
Before training	0.428	0.125	0.394	0.214	0.202	0.643
After training	0.636	0.698	0.243	0.308	0.314	0.726
Domain 3: Outcome						
Before training	0.572	0.588	0.278	0.510	0.774	0.647
After training	0.851	0.776	0.899	0.899	0.683	0.947
Domain 4: Analysis						
Before training	0.493	0.635	0.085	0.108	0.145	0.629
After training	0.606	0.740	0.222	0.413	−0.022	0.802
Overall						
Before training	0.562	0.479	0.216	−0.313	−0.256	0.694
After training	0.711	0.713	0.423	0.537	0.392	0.893

**Table 2 jcm-12-01976-t002:** Mean pairwise raw agreement, mean pairwise AC_1_ and mean Cohen’s κ before and after training for the domain-specific and overall ROB rating.

	Mean Raw Agreement		Mean Pairwise AC_1_		Mean Cohen’s κ	
	Before Training	After Training	Before Training	After Training	Before Training	After Training
Domain 1: Participants	0.530	0.615	0.357	0.464	0.167	0.396
Domain 2: Predictors	0.465	0.494	0.284	0.297	0.019	0.171
Domain 3: Outcome	0.637	0.809	0.534	0.776	0.183	0.310
Domain 4: Analysis	0.397	0.524	0.142	0.298	0.134	0.287
Overall	0.377	0.612	0.098	0.474	0.132	0.261

## Data Availability

Most of the data is contained within the article and Appendix A. Data for the pairwise agreement are available on request from the corresponding author.

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
