# Peer review of "Inter-Rater Agreement in Assessing Risk of Bias in Melanoma Prediction Studies Using the Prediction Model Risk of Bias Assessment Tool (PROBAST): Results from a Controlled Experiment on the Effect of Specific Rater Training"

_jcm, 2023, doi:10.3390/jcm12051976_

Round 1

Reviewer 1 Report

This study examined the IRR of Prediction Model Risk of Bias Assessment tool and the effect of training sessions on IRR among melanoma prediction studies. The topic is of importance and the manuscript is well-written.  I only have minor comments below:

The authors stated the importance of risk of bias and its assessment tool but it’s unclear why the authors chose melanoma prediction models to examine. 

Line 90-91, page 2. The author mentioned that the six raters were at different levels of experience. What does this mean? Did you mean experience of risk of bias assessment?

Line 202-205, Page 7. Is there any explanation for why the domain “analysis” and overall ROB had low agreement? During the training course, did the authors discuss certain domains more?

In the discussion and Appendix, the authors mentioned the rationale of using Gwet AC instead of Kappa to estimate the IRR.  In Section 33.4, Cohen’s K is also provided. As this is not a methodological research article, no further interpretation or analysis is provided in this section. Therefore, the authors may put the additional results of Cohen's K in supplementary instead.

Reviewer 2 Report

In Kaiser’s et al. article “Inter-rater Agreement in Assessing Risk of Bias in Melanoma Prediction Studies Using the Prediction Model Risk of Bias Assessment Tool (PROBAST): Results from a Controlled Experiment With Six Raters”, she explores the inter-rater reliability (IRR) of the PROBAST, which was rated by six raters. Overall, the article is well written, to the point, and provides answers to a very relevant question in the field of ROB: is the PROBAST a valid instrument to assess ROB in prediction research. I have the following minor suggestions (no major concerns):

Title: it might be a matter of taste, but I feel the present title doesn’t apply to the article. Something like “Effect of training on the inter-rater agreement on the Prediction Model Risk of Bias Assessment Tool (PROBAST): a case study in melanoma prediction studies [with six rater]” Text between brackets could be removed.

Abstract:

1)      “Gwet’s AC1 was used to quantify the pairwise and multi-rater IRR” . Perhaps also add that you’ve used other methods as well?

2)      The abstract is not as clearly written as the article itself (perhaps it was rewritten?). I don’t feel the urgency I felt reading the first paragraph of the discussion. The casual reader will read only the abstract to decide whether the article is interesting enough to read. This is the place to make bold, and to the point comments. At least add a clear sentence like “Without targeted guidance, the reliability of the PROBAST instrument is low (from conclusion)”  

Under 2.2 (“ROB assessment using the PROBAST”), the authors describe the PROBAST, and its four domains. I have three suggestions;

1)      Presumably, the casual reader will not know the PROBAST. It would help to add general information to the introduction (how the PROBAST was developed, why it was developed, and how the authors of the PROBAST define ‘ROB’ [which is not that very straightforward in prediction research])

2)      In paragraph 2.2, the authors could then very limited reiterate this, and provide a more detailed overview of the four domains: i.e. the number of signaling questions (SQ), how low/unclear/high ROB per SQ translate to overall ROB of each domain. As the authors know, the final two domains contain more SQ (esp. domain 4), still, only one SQ answered as ‘high’ will result in overall high ROB of that domain. This should be stated somewhere in the article, as it will explain the overall high ROB of prediction studies as assessed with the PROBAST.

3)      This refers to this sentence: PROBAST instrument provided in [12] was used. This references refers to the PROBAST website, which contains the short and elaboration. Perhaps add this information, so the reader will know how ‘informed’ the raters are. Also please add any previous experience with the PROBAST of each author.

Under 2.3 (“Rating process and training”) the authors write: In case of sustained disagreement, two independent referees (A.B.P. and W.U.) made the final decisions. I’m curious: for how many instances was this the case? Providing this information will help the reader to understand the complexity of some SQ (apparently, even after two meetings of four hours, there was still disagreement).

Under 2.4 (“Statistical analysis”), please expand a little on the Gwet’s AC1 statistic. I’m not familiar with this statistic, and how it correlates to Cohen’s Kappa. This will also help the reader to understand section 3.4 and table 2.  

Under 3.1 (“study characteristics”)

1)      I’m not sure how to interpret the first sentence. PROBAST does not provide an ‘overall ROB’, it provides an overall ROB per domain. I would suggest to present data for the four domains separately.  

2)      Please provide a figure (e.g. bar plot) of the ROB per domain, as also suggested by the PROBAST authors.

Regarding the discussion (which is concise, to the point, and well written):

1)      “Our results showed that without guidance or specific training the IRR of the PROBAST instrument is low.” Please add a consequence / interpretation: the casual reader might think ‘low IRR = good’. Adding something like ‘… is low, meaning that the tool is not valid for ROB assessment without proper training’. An explanation for this low IRR is given in the second paragraph (“This clearly demonstrates the subjectivity of the PROBAST instrument and its rater dependency”).  

2)      “This is the first study assessing the reliability of the PROBAST instrument.” This is incorrect, there is one other study:

a.       https://pubmed.ncbi.nlm.nih.gov/34175377/

This is in line with your results. Please include in your discussion.

3)      “explanations, like those available for PROBAST,…”, perhaps also add ref #13.

4)      Perhaps also expand on the generalizability of your findings to other fields than melanoma research. I would presume the findings to be generalizable, but expanding a bit on that might guide the reader.

The conclusion:

1)      “Without targeted guidance, the reliability of the PROBAST instrument is low, …”  I would use ‘validity’ instead of reliability.

2)      “… which has negative implications for the consistency of research synthesis from prediction studies and for subsequent decision-making.” I had to re-read this this sentence thrice. In the conclusion, I would be concise and clear. If you believe the PROBAST is invalid for ROB assessment without proper training or experience (as you point out in the discussion above), write that down.  
